# Cold Atoms in *U*(3) Gauge Potentials

**Ipsita Mandal** [1,*] **and Atri Bhattacharya** [2]

1    Perimeter Institute for Theoretical Physics, 31 Caroline St. N., Waterloo, ON N2L 2Y5, Canada
2    IFPA, AGO Department, University of Liège, Sart Tilman, 4000 Liège, Belgium; A.Bhattacharya@ulg.ac.be
*    Correspondence: imandal@pitp.ca

**Abstract:** We explore the effects of artificial *U*(3) gauge potentials on ultracold atoms. We study background gauge fields with both non-constant and constant Wilson loops around plaquettes, obtaining the energy spectra in each case. The scenario of metal–insulator transition for irrational fluxes is also examined. Finally, we discuss the effect of such a gauge potential on the superfluid–insulator transition for bosonic ultracold atoms.

**Keywords:** cold atoms; non-Abelian; *U*(3)

## 1. Introduction

The study of ultracold atoms in optical lattices has emerged to be a subject of great interest in recent years, opening up the possibilities of synthesising gauge fields capable of coupling to neutral atoms. This is in a vein similar to how electromagnetic fields couple to charged matter, for instance, or how $SU(2)$ and $SU(3)$ fields couple to fundamental particles in high-energy physics [1–10]. The effects of these artificial Abelian and non-Abelian "magnetic fields" can subsequently be studied in experiments designed to realise these magnetic fields. Over the years, several innovative techniques to achieve this have been suggested. One such procedure involves rotating the atoms in a trap [2,11]. More sophisticated methods involve atoms in optical lattices, making use of laser-assisted tunnelling and lattice tilting (acceleration) [12–14], laser methods employing dark states [15,16], two-photon dressing by laser fields [17,18], lattice rotations [19–22], or immersion of atoms in a lattice within a rotating Bose–Einstein condensate [23]. Further, in a recent work [24], the authors have proposed a two-tripod scheme to generate artificial $U(3)$ gauge fields. Observations in these experiments are expected to show particularly conspicuous features, like the fractal "Hofstadter butterfly" spectrum [25] and the "Escher staircase" [13] in single-particle spectra, vortex formation [2,19,26], quantum Hall effects [14,21,27,28], as well as other quantum correlated liquids [29].

A novel scheme to generate artificial *Abelian* "magnetic" fields was proposed in the work by Jaksch and Zoller [12]. This involves the coherent transfer of atoms between two different internal states by making use of Raman lasers. Later, by making use of laser tunneling between $N$ distinct internal states of an atom, this scheme was generalised to mimic artifical non-Abelian "magnetic" fields by Osterloh et al [30]. In addition, an alternative method—employing dark states—has also been discussed [24,31]. In such a scenario, one employs atoms with multiple internal states, dubbed "flavours". The gauge potentials that can be realized by the application of laser-assisted, non-uniform, and state-dependent tunnelling and coherent transfer between internal states, can practically allow for a unitary matrix transformation in the space of these internal states, corresponding to $U(N)$ or $SU(N)$. In such a non-Abelian $U(2)$ potential, a moth-like structure [30] emerges for the single-particle spectrum, which is characterized by numerous tiny gaps. Several

other works involve studies of non-trivial quantum transport properties [32], integer quantum Hall effect for cold atoms [28], spatial patterns in optical lattices [26], modifications of the Landau levels [33], and quantum atom optics [34,35]. An $SU(3)$ topological insulator has been constructed for a non-interacting quadratic Hamiltonian [36]. In the context of an interacting system with three-component bosons, the Mott phase in the presence of "$SU(3)$ spin-orbit coupling" has been shown to exhibit $SU(3)$ spin spiral textures in the ground state, both for the one-dimensional chain and the square lattice [37].

However, Goldman et al [38] have pointed out that the $U(2)$ gauge potentials proposed earlier [30] are characterized by non-constant Wilson loops and that the features characterizing the Hofstadter "moth" are a consequence of this spatial dependence of the Wilson loop, rather than the non-Abelian nature of the potential. They have emphasised that the moth-like spectrum can also be found in the standard Abelian case when the gauge potential is chosen such that the Wilson loop is proportional to the spatial coordinate.

In this work, we investigate whether features similar to those discussed in the literature for $U(2)$ gauge potentials also reveal themselves in artificial $U(3)$ gauge potentials on ultracold atoms. This builds upon existing results in the literature for $U(2)$ potentials and may be viewed as a stepping stone toward the generalisation of such features for arbitrary $(S)U(N)$ gauge potentials.

Our paper is organised as follows. Section 2 describes the necessary theoretical set-up. In Section 3, we consider background gauge fields with non-constant Wilson loops [30]. The spectra for both rational and irrational fluxes are discussed. The scenario of metal–insulator transition for irrational fluxes is also examined in Section 3.2. Section 4 is devoted to systems subjected to a gauge potential with a constant [38] Wilson loop. Lastly, in Section 4.2, we study the effect of such a gauge potential on the Mott insulator to superfluid transition for bosonic ultracold atoms for rational fluxes. We conclude with a summary and an outlook for related future work in Section 5.

## 2. Review of Artificial Gauge Potentials in Optical Lattices

In this Section we review the theoretical framework for studying a system of non-interacting fermionic atoms with *j* flavours. We assume that the atoms are trapped in a 2D optical square lattice of lattice-spacing *a* with sites at $(x = m\,a, y = n\,a)$, where $n, m$ are integers. Without loss of generality, we will set $a = 1$ in all subsequent discussions. When the optical potential is strong, the tight-binding approximation holds and the Hamiltonian is given by

$$H = \sum_{m,n} \left( t_x\, \Psi_{m+1,n}^{\dagger}\, U_x \Psi_{m,n}\, +\, t_y\, \Psi_{m,n+1}^{\dagger}\, U_y\, \Psi_{m,n} \right) + h.c., \tag{1}$$

where $U_x$ and $U_y$ are the tunnelling matrices (operators), belonging to the $U(N)$ group, along the $x$ and $y$ directions respectively. Also, $t_x$ and $t_y$ represent the corresponding tunnelling amplitudes, and each of the $\Psi_{m,n}^{\dagger}$'s is a $j$-component fermion creation operator at the site $(m, n)$. The tunnelling operators are related to the non-Abelian gauge potential according to $U_x = e^{iA_x}$ and $U_y = e^{iA_y}$. Throughout this work, we will impose periodic boundary conditions in both $x$ and $y$ directions.

In the presence of the gauge potential, the atoms performing a loop around a plaquette undergo the unitary transformation

$$U = U_x\, U_y(m+1)\, U_x^{\dagger}\, U_y^{\dagger}(m)\,, \tag{2}$$

where we are considering the case that $U_x$ is position-independent, whereas $U_y$ depends on the $x$-coordinate. Noting that the gauge potential (and hence the Hamiltonian) is independent of the $y$-coordinate, the three-component eigenfunction can be written as

$$\Psi(m, n) \equiv e^{ik_y n} \begin{pmatrix} a_m \\ b_m \\ c_m \end{pmatrix}, \tag{3}$$

such that $H \ket{\Psi} = E \ket{\Psi}$.

The Wilson loop defined by

$$W = \mathrm{Tr}\left[\, U_x(m+1)\, U_y(m+1)\, U_x^\dagger(m)\, U_y^\dagger(m)\,\right], \tag{4}$$

is a gauge-invariant quantity and can be used to distinguish whether the system is in the "genuine" Abelian or non-Abelian regime. For $|W| = 3$, the system is in the Abelian regime according to the criteria by Goldman et al. [38].

## 3. *U(3)* Gauge Potential with Non-Constant Wilson Loop

In this section, we consider the $U(3)$ gauge potential

$$
\begin{aligned}
A_x &= \frac{4\pi}{3\sqrt{3}} \begin{pmatrix} 0 & -i & i \\ i & 0 & -i \\ -i & i & 0 \end{pmatrix}, \\
A_y &= -2\pi m \, \mathrm{diag}(\alpha_1, \alpha_2, \alpha_3), \\
A_z &= 0,
\end{aligned}
\tag{5}
$$

where $A_x$ is proportional to the linear combination $(\lambda_2 - \lambda_5 + \lambda_7)$ of the Gell-Mann matrices for $SU(3)$. In order to realize such a potential one may consider the method elaborated by Osterloh et al. [30].

The tunnelling operators corresponding to the above non-Abelian gauge potentials are given by the following $3 \times 3$ unitary matrices

$$
\begin{aligned}
U_x &= \begin{pmatrix} 0 & 0 & 1 \\ 1 & 0 & 0 \\ 0 & 1 & 0 \end{pmatrix}, \\
U_y &= \mathrm{diag}\left(e^{-i2\pi\alpha_1 m},\, e^{-i2\pi\alpha_2 m},\, e^{-i2\pi\alpha_3 m}\right).
\end{aligned}
\tag{6}
$$

From Equation (4), we find $W = e^{2\pi i\{m\alpha_1 - (m+1)\alpha_3\}} + e^{2\pi i\{m\alpha_1 - (m+1)\alpha_3\}} + e^{2\pi i\{m\alpha_3 - (m+1)\alpha_2\}}$, which is position-dependent for generic values of $\alpha_1, \alpha_2, \alpha_3$, and hence we expect a moth-like (rather than butterfly-like) structure [38]. For $\alpha_1 = \alpha_2 = \alpha_3$, $|W| = 3$ and we are then in the Abelian regime where the fractal "Hofstadter butterfly" is expected to show up with $q_1 (= q_2 = q_3)$ triply-degenerate bands.

### 3.1. Spectrum for Rational Fluxes

For the case of rational $\alpha_i$s such that

$$\alpha_i = p_i / q_i \quad (\text{for } i \in \{1,2,3\} \ \& \ \{p_i, q_i\} \in \mathbb{Z}), \tag{7}$$

the system is periodic in the $x$-direction, with periodicity $Q$, where $Q$ is equal to the least common multiple of $\{q_1, q_2, q_3\}$. The recursive eigenvalue equations are

$$
\begin{aligned}
b_{m-1} + U_m\, a_m + c_{m+1} &= \frac{E}{t_x}\, a_m, \\
c_{m-1} + V_m\, b_m + a_{m+1} &= \frac{E}{t_x}\, b_m, \\
a_{m-1} + W_m\, c_m + b_{m+1} &= \frac{E}{t_x}\, c_m,
\end{aligned}
\tag{8}
$$

where

$$
\begin{aligned}
U_m &= 2\,r\cos(2\pi m\alpha_1 - k_y), \quad V_m = 2\,r\cos(2\pi m\alpha_2 - k_y),\\
W_m &= 2\,r\cos(2\pi m\alpha_3 - k_y), \quad r = t_y/t_x.
\end{aligned}
\tag{9}
$$

Since the Hamiltonian $H$ commutes with the translation operator defined by $\mathcal{T}_x^Q f(m,n) = f(m+Q,n)$, we can apply Bloch's theorem in the $x$-direction

$$
\begin{pmatrix} a_{m+Q} \\ c_{m+Q} \\ b_{m+Q} \end{pmatrix} = e^{ik_x Q} \begin{pmatrix} a_m \\ c_m \\ b_m \end{pmatrix}.
\tag{10}
$$

Hence, in the first Brillouin zone, $k_x \in [0, \frac{2\pi}{Q}]$ and $k_y \in [0, 2\pi]$, and we need to solve the $3Q \times 3Q$ eigenvalue problem:

$$
\begin{pmatrix}
U_1 & 0 & 0 & 0 & 0 & 1 & 0 & 0 & 0 & . & . & 0 & 0 & 0 & 0 & e^{-ik_xQ} & 0 \\
0 & V_1 & 0 & 1 & 0 & 0 & 0 & 0 & 0 & . & . & 0 & 0 & 0 & 0 & 0 & e^{-ik_xQ} \\
0 & 0 & W_1 & 0 & 1 & 0 & 0 & 0 & 0 & . & . & 0 & 0 & 0 & e^{-ik_xQ} & 0 & 0 \\
0 & 1 & 0 & U_2 & 0 & 0 & 0 & 0 & 1 & . & . & 0 & 0 & 0 & 0 & 0 & 0 \\
0 & 0 & 1 & 0 & V_2 & 0 & 1 & 0 & 0 & . & . & 0 & 0 & 0 & 0 & 0 & 0 \\
1 & 0 & 0 & 0 & 0 & W_2 & 0 & 1 & 0 & . & . & 0 & 0 & 0 & 0 & 0 & 0 \\
. & . & . & . & . & . & . & . & . & & & . & . & . & . & . & . \\
. & . & . & . & . & . & . & . & . & & & . & . & . & . & . & . \\
0 & 0 & e^{ik_xQ} & 0 & 0 & 0 & 0 & 0 & 0 & . & . & 0 & 1 & 0 & U_Q & 0 & 0 \\
e^{ik_xQ} & 0 & 0 & 0 & 0 & 0 & 0 & 0 & 0 & . & . & 0 & 0 & 1 & 0 & V_Q & 0 \\
0 & e^{ik_xQ} & 0 & 0 & 0 & 0 & 0 & 0 & 0 & . & . & 1 & 0 & 0 & 0 & 0 & W_Q
\end{pmatrix}
\begin{pmatrix} a_1 \\ b_1 \\ c_1 \\ a_2 \\ b_2 \\ c_2 \\ . \\ . \\ a_Q \\ b_Q \\ c_Q \end{pmatrix}
= \frac{E}{t_x}
\begin{pmatrix} a_1 \\ b_1 \\ c_1 \\ a_2 \\ b_2 \\ c_2 \\ . \\ . \\ a_Q \\ b_Q \\ c_Q \end{pmatrix}.
\tag{11}
$$

This matrix equation can be decoupled into three independent equations:

$$
\begin{pmatrix}
U_1 & 1 & 0 & 0 & 0 & 0 & 0 & 0 & 0 & . & . & 0 & 0 & 0 & 0 & 0 & e^{-ik_xQ} \\
1 & W_2 & 1 & 0 & 0 & 0 & 0 & 0 & 0 & . & . & 0 & 0 & 0 & 0 & 0 & 0 \\
0 & 1 & V_3 & 1 & 0 & 0 & 0 & 0 & 0 & . & . & 0 & 0 & 0 & 0 & 0 & 0 \\
0 & 0 & 1 & U_4 & 1 & 0 & 0 & 0 & 0 & . & . & 0 & 0 & 0 & 0 & 0 & 0 \\
0 & 0 & 0 & 1 & W_5 & 1 & 0 & 0 & 0 & . & . & 0 & 0 & 0 & 0 & 0 & 0 \\
0 & 0 & 0 & 0 & 1 & V_6 & 1 & 0 & 0 & . & . & 0 & 0 & 0 & 0 & 0 & 0 \\
. & . & . & . & . & . & . & . & . & & & . & . & . & . & . & . \\
. & . & . & . & . & . & . & . & . & & & . & . & . & . & . & . \\
0 & 0 & 0 & 0 & 0 & 0 & 0 & 0 & 0 & . & . & 0 & 0 & 1 & U_{Q-2} & 1 & 0 \\
0 & 0 & 0 & 0 & 0 & 0 & 0 & 0 & 0 & . & . & 0 & 0 & 0 & 1 & W_{Q-1} & 1 \\
e^{ik_xQ} & 0 & 0 & 0 & 0 & 0 & 0 & 0 & 0 & . & . & 0 & 0 & 0 & 0 & 1 & V_Q
\end{pmatrix}
\begin{pmatrix} a_1 \\ c_2 \\ b_3 \\ a_4 \\ c_5 \\ b_6 \\ . \\ . \\ a_{Q-2} \\ c_{Q-1} \\ b_Q \end{pmatrix}
= \frac{E_1}{t_x}
\begin{pmatrix} a_1 \\ c_2 \\ b_3 \\ a_4 \\ c_5 \\ b_6 \\ . \\ . \\ a_{Q-2} \\ c_{Q-1} \\ b_Q \end{pmatrix},
\tag{12}
$$

$$
\begin{pmatrix}
V_1 & 1 & 0 & 0 & 0 & 0 & 0 & 0 & 0 & . & . & 0 & 0 & 0 & 0 & 0 & e^{-ik_xQ} \\
1 & U_2 & 1 & 0 & 0 & 0 & 0 & 0 & 0 & . & . & 0 & 0 & 0 & 0 & 0 & 0 \\
0 & 1 & W_3 & 1 & 0 & 0 & 0 & 0 & 0 & . & . & 0 & 0 & 0 & 0 & 0 & 0 \\
0 & 0 & 1 & V_4 & 1 & 0 & 0 & 0 & 0 & . & . & 0 & 0 & 0 & 0 & 0 & 0 \\
0 & 0 & 0 & 1 & U_5 & 1 & 0 & 0 & 0 & . & . & 0 & 0 & 0 & 0 & 0 & 0 \\
0 & 0 & 0 & 0 & 1 & W_6 & 1 & 0 & 0 & . & . & 0 & 0 & 0 & 0 & 0 & 0 \\
. & . & . & . & . & . & . & . & . & & & . & . & . & . & . & . \\
. & . & . & . & . & . & . & . & . & & & . & . & . & . & . & . \\
0 & 0 & 0 & 0 & 0 & 0 & 0 & 0 & 0 & . & . & 0 & 0 & 1 & V_{Q-2} & 1 & 0 \\
0 & 0 & 0 & 0 & 0 & 0 & 0 & 0 & 0 & . & . & 0 & 0 & 0 & 1 & U_{Q-1} & 1 \\
e^{ik_xQ} & 0 & 0 & 0 & 0 & 0 & 0 & 0 & 0 & . & . & 0 & 0 & 0 & 0 & 1 & W_Q
\end{pmatrix}
\begin{pmatrix} b_1 \\ a_2 \\ c_3 \\ b_4 \\ a_5 \\ c_6 \\ . \\ . \\ b_{Q-2} \\ a_{Q-1} \\ c_Q \end{pmatrix}
= \frac{E_2}{t_x}
\begin{pmatrix} b_1 \\ a_2 \\ c_3 \\ b_4 \\ a_5 \\ c_6 \\ . \\ . \\ b_{Q-2} \\ a_{Q-1} \\ c_Q \end{pmatrix},
\tag{13}
$$

$$\begin{pmatrix} W_1 & 1 & 0 & 0 & 0 & 0 & 0 & 0 & 0 & . & . & 0 & 0 & 0 & 0 & 0 & e^{-ik_xQ} \\ 1 & V_2 & 1 & 0 & 0 & 0 & 0 & 0 & 0 & . & . & 0 & 0 & 0 & 0 & 0 & 0 \\ 0 & 1 & U_3 & 1 & 0 & 0 & 0 & 0 & 0 & . & . & 0 & 0 & 0 & 0 & 0 & 0 \\ 0 & 0 & 1 & W_4 & 1 & 0 & 0 & 0 & 0 & . & . & 0 & 0 & 0 & 0 & 0 & 0 \\ 0 & 0 & 0 & 1 & V_5 & 1 & 0 & 0 & 0 & . & . & 0 & 0 & 0 & 0 & 0 & 0 \\ 0 & 0 & 0 & 0 & 1 & U_6 & 1 & 0 & 0 & . & . & 0 & 0 & 0 & 0 & 0 & 0 \\ . & . & . & . & . & . & . & . & . & . & . & . & . & . & . & . & . \\ . & . & . & . & . & . & . & . & . & . & . & . & . & . & . & . & . \\ . & . & . & . & . & . & . & . & . & . & . & . & . & . & . & . & . \\ 0 & 0 & 0 & 0 & 0 & 0 & 0 & 0 & 0 & . & . & 0 & 0 & 1 & W_{Q-2} & 1 & 0 \\ 0 & 0 & 0 & 0 & 0 & 0 & 0 & 0 & 0 & . & . & 0 & 0 & 0 & 1 & V_{Q-1} & 1 \\ e^{ik_xQ} & 0 & 0 & 0 & 0 & 0 & 0 & 0 & 0 & . & . & 0 & 0 & 0 & 0 & 1 & U_Q \end{pmatrix} \begin{pmatrix} c_1 \\ b_2 \\ a_3 \\ b_4 \\ a_5 \\ c_6 \\ . \\ . \\ . \\ c_{Q-2} \\ b_{Q-1} \\ a_Q \end{pmatrix} = \frac{E_3}{t_x} \begin{pmatrix} c_1 \\ b_2 \\ a_3 \\ b_4 \\ a_5 \\ c_6 \\ . \\ . \\ . \\ c_{Q-2} \\ b_{Q-1} \\ a_Q \end{pmatrix},$$

such that the full set of eigenvalues is the union of the eigenvalues $(E_1, E_2, E_3)$ obtained for the three decoupled systems. Figure 1 shows the plots of these energy eigenvalues as functions of $\alpha_1$ for $\alpha_2 = 2/3$, $\alpha_3 = 1/2$ and $r = 1$. The three plots, from left to right, correspond to $k_y = 0, \pi/3, \pi/2$ respectively. We have checked that the features of the plots remain unchanged irrespective of whether the horizontal axis is chosen as $\alpha_1$, $\alpha_2$ or $\alpha_3$, whilst keeping the other two $\alpha_i$s fixed.

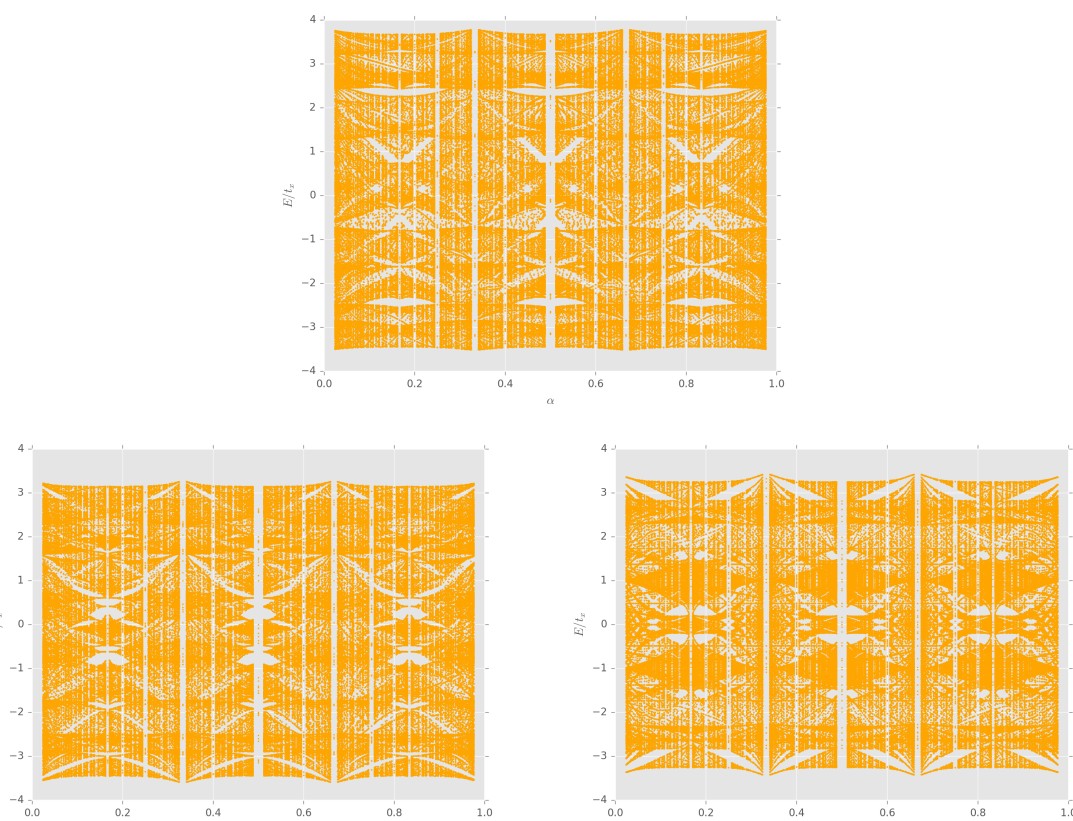

**Figure 1.** The energy eigenvalues of the system depicted by Equation (11) for $\alpha_2 = 2/3$, $\alpha_3 = 1/2$ and $r = 1$ as a function of $\alpha_1$. The three plots correspond to $k_y = 0$ (top), $\pi/3$ (bottom left) and $\pi/2$ (bottom right).

### 3.2. Metal–Insulator Transition for Irrational Flux

The Hofstadter system [25] undergoes metal–insulator transitions for irrational values of flux and the spectra do not depend on $k_y$. For instance, let us assume that $\alpha_1 = \frac{\sqrt{5}-1}{2}$. We will approximate this irrational number by the rational approximation 89/144. Figure 2 shows the plot of the energy

eigenvalues from Equations (12)–(14) as a function of $k_y$ for $\alpha_1 = 89/144$, $\alpha_2 = 2/3$, $\alpha_3 = 1/2$ and $r = 1$. The Abelian case corresponding to $\alpha_1 = \alpha_2 = \alpha_3 = 89/144$ has also been shown, which shows bands with no variation along $k_y$. Also in Figure 3, we show how the minimum energy states for $k_y = (0, \pi/2)$ localizes with increasing $r$.

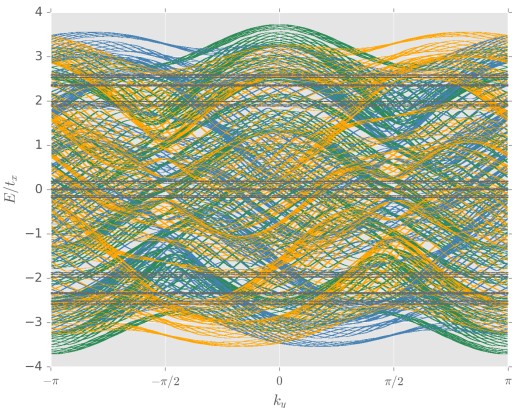

**Figure 2.** Energy spectrum $E = (E_1, E_2, E_3)$ from Equations (12)–(14), as a function of $k_y$ for $\alpha_1 = 89/144$, $\alpha_2 = 2/3$, $\alpha_3 = 1/2$ and $r = 1$. $E_1, E_2$ and $E_3$ have been plotted in green, blue and orange respectively. The Abelian case corresponding to $\alpha_1 = \alpha_2 = \alpha_3 = 89/144$ has also been plotted in grey, which shows bands with no variation with $k_y$.

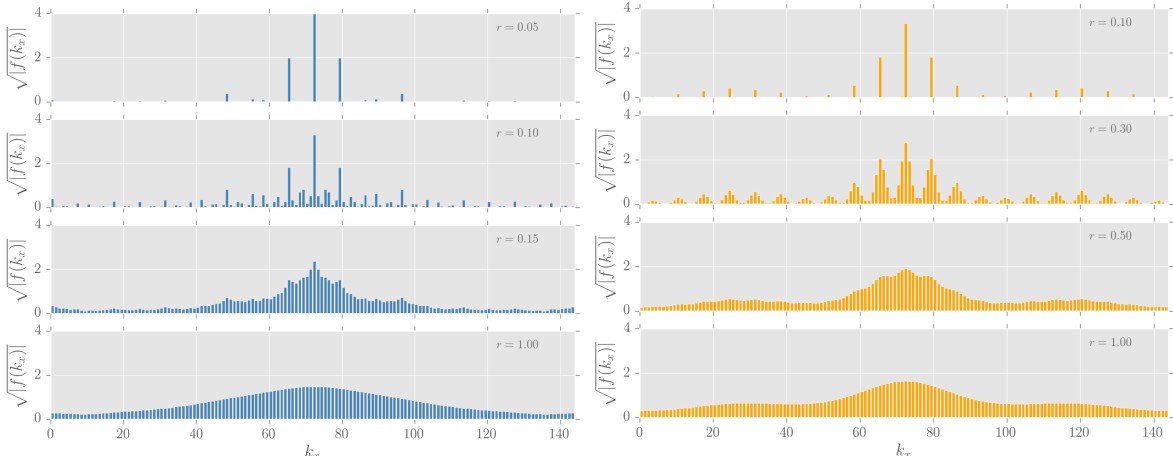

**Figure 3.** The behaviour of the system depicted by Equation (10) for irrational flux, captured by plotting the square root of the modulus of the Fourier transform of the wavefunction $\left( \sqrt{|f(k_x)|} \right)$ for the state with minimum energy when $\alpha_1 = 89/144$, $\alpha_2 = 2/3$, $\alpha_3 = 1/2$ as a function of $k_x$ with $k_y = 0\,(\pi/2)$ for left (right) panel. The four figures in each panel, from top to bottom, show how the state localizes with increasing $r$.

If we consider the case of $A_y = -2\pi m \operatorname{diag}(\alpha_1, \alpha_2 + 1/2, \alpha_3)$ such that $U_y = \operatorname{diag}(e^{-i2\pi\alpha_1 m}, -e^{-i2\pi\alpha_2 m}, e^{-i2\pi\alpha_3 m})$ (other choices remaining the same as in Equation (5)), then for $E = 0$, an irrational value of $\alpha_1$ and $\alpha_2 = \alpha_3 = 0$, the recursive equations reduce to

$$a(m-3) + a(m+3) + 2\gamma\cos\left(2\pi m\alpha_1 - k_y\right) a(m) = 0,$$

$$b(m) = \frac{a(m-2) + 2r(-1)^m \cos k_y\, a(m+1)}{-1 + 4r^2 \cos^2 k_y},$$

$$c(m) = \frac{a(m-1) - b(m+1)}{2r(-1)^m \cos k_y},$$

$$\gamma = -r\left(1 - 4r^2\cos^2 k_y\right). \tag{14}$$

The $a(m)$ equation is uncoupled and has a similar structure to the Harper equation for the Abelian case. This leads us to infer that there is a metal–insulator transition at $|\gamma| = 1$, such that $|\gamma| < 1$ corresponds to extended states, while $|\gamma| > 1$ characterises localized states. These two phases are shown in Figure 4.

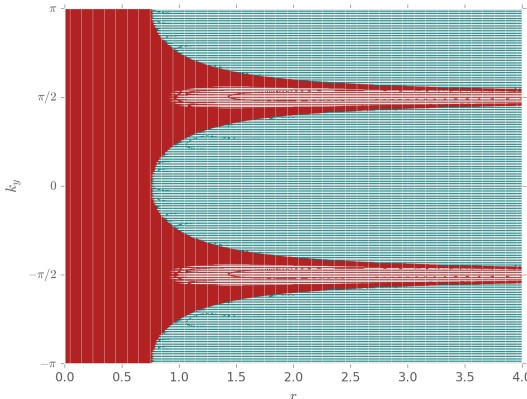

**Figure 4.** The metallic (red) and insulating phases (blue) for the system from Equation (14) in the $k_y - r$ plane.

## 4. $U(3)$ Gauge Potential with Constant Wilson Loop

In this section, we study the effect of the $U(3)$ gauge potential given by

$$A_x = \frac{4\pi}{3\sqrt{3}} \begin{pmatrix} 0 & -i & i \\ i & 0 & -i \\ -i & i & 0 \end{pmatrix},$$

$$A_y = 2\pi\alpha m + \frac{\pi}{\sqrt{3}} \begin{pmatrix} 0 & -i & -i \\ i & 0 & -i \\ i & i & 0 \end{pmatrix},$$

$$A_z = 0, \tag{15}$$

where $A_x$ is the same as in Equation (5) but $A_y$ now is proportional to the linear combination $(\lambda_2 + \lambda_5 + \lambda_7)$ of the Gell-Mann matrices for $SU(3)$. The tunnelling operators in this case correspond to the following unitary matrices:

$$U_x = \begin{pmatrix} 0 & 0 & 1 \\ 1 & 0 & 0 \\ 0 & 1 & 0 \end{pmatrix}, \quad U_y = -\frac{e^{i2\pi\alpha m}}{3} \begin{pmatrix} 1 & 2 & -2 \\ 2 & 1 & 2 \\ -2 & 2 & 1 \end{pmatrix}. \tag{16}$$

Here, Equation (4) gives us $|W| = 5/9$, which is position-independent and hence we expect a modified butterfly structure.

### 4.1. Spectrum for Rational Flux

For $\alpha = P/Q$, writing the wave-functions in terms of Bloch functions using the same notation as in Equation (10), we arrive at the recursive equations given by

$$
\begin{aligned}
b_{m-1} + \tilde{U}_m\left(a_m + 2b_m - 2c_m\right) + c_{m+1} &= \frac{E}{t_x}\,a_m, \\
c_{m-1} + \tilde{U}_m\left(2a_m + b_m + 2c_m\right) + a_{m+1} &= \frac{E}{t_x}\,b_m, \\
a_{m-1} + \tilde{U}_m\left(-2a_m + 2b_m + c_m\right) + b_{m+1} &= \frac{E}{t_x}\,c_m,
\end{aligned}
\tag{17}
$$

where

$$
\tilde{U}_m = -2\,r\cos(2\pi m\alpha + k_y)\,, \quad r = t_y/t_x\,.
\tag{18}
$$

This case involves solving a $3Q \times 3Q$ eigenvalue problem given by

$$
\begin{pmatrix}
\tilde{U}_1 & 2\tilde{U}_1 & -2\tilde{U}_1 & 0 & 0 & 1 & 0 & 0 & 0 & . & . & 0 & 0 & 0 & 0 & e^{-ik_xQ} & 0 \\
2\tilde{U}_1 & \tilde{U}_1 & 2\tilde{U}_1 & 1 & 0 & 0 & 0 & 0 & 0 & . & . & 0 & 0 & 0 & 0 & 0 & e^{-ik_xQ} \\
-2\tilde{U}_1 & 2\tilde{U}_1 & \tilde{U}_1 & 0 & 1 & 0 & 0 & 0 & 0 & . & . & 0 & 0 & 0 & e^{-ik_xQ} & 0 & 0 \\
0 & 1 & 0 & \tilde{U}_2 & 2\tilde{U}_2 & -2\tilde{U}_2 & 0 & 0 & 1 & . & . & 0 & 0 & 0 & 0 & 0 & 0 \\
0 & 0 & 1 & 2\tilde{U}_2 & \tilde{U}_2 & 2\tilde{U}_2 & 1 & 0 & 0 & . & . & 0 & 0 & 0 & 0 & 0 & 0 \\
1 & 0 & 0 & -2\tilde{U}_2 & 2\tilde{U}_2 & \tilde{U}_2 & 0 & 1 & 0 & . & . & 0 & 0 & 0 & 0 & 0 & 0 \\
. & . & . & & & & . & . & . & . & . & . & & . & & . \\
. & . & . & & & & . & . & . & . & . & . & & . & & . \\
0 & 0 & e^{ik_xQ} & 0 & 0 & 0 & 0 & 0 & 0 & . & . & 0 & 1 & 0 & \tilde{U}_Q & 2\tilde{U}_Q & -2\tilde{U}_Q \\
e^{ik_xQ} & 0 & 0 & 0 & 0 & 0 & 0 & 0 & 0 & . & . & 0 & 0 & 1 & 2\tilde{U}_Q & \tilde{U}_Q & 2\tilde{U}_Q \\
0 & e^{ik_xQ} & 0 & 0 & 0 & 0 & 0 & 0 & 0 & . & . & 1 & 0 & 0 & -2\tilde{U}_Q & 2\tilde{U}_Q & \tilde{U}_Q
\end{pmatrix}
\begin{pmatrix}
a_1 \\ b_1 \\ c_1 \\ a_2 \\ b_2 \\ c_2 \\ . \\ . \\ a_Q \\ b_Q \\ c_Q
\end{pmatrix}
= \frac{E}{t_x}
\begin{pmatrix}
a_1 \\ b_1 \\ c_1 \\ a_2 \\ b_2 \\ c_2 \\ . \\ . \\ a_Q \\ b_Q \\ c_Q
\end{pmatrix}.
\tag{19}
$$

In Figure 5, the energy eigenvalues (with $r = 1$) have been plotted as a function of (i) $\alpha$ in the left panel, and (ii) $k_y$ for $\alpha = 3/5$ in the right panel.

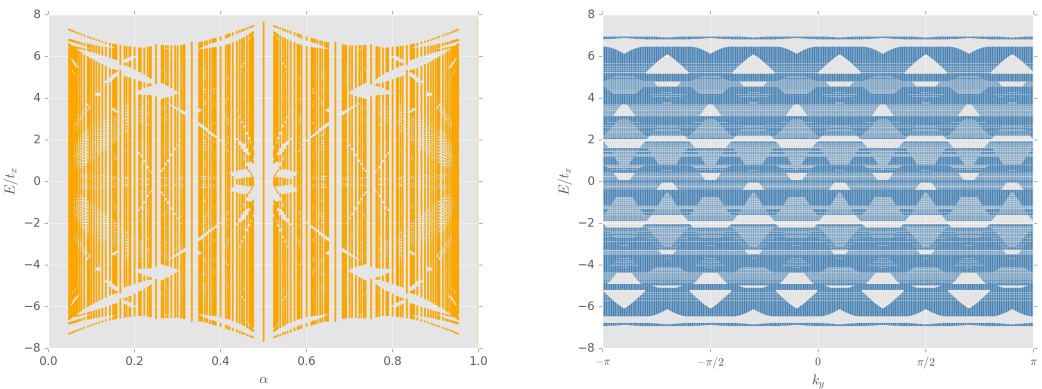

**Figure 5.** Energy spectrum from Equation (19) for $r = 1$ as a function of (i) $\alpha$ in the left panel, and (ii) $k_y$ at $\alpha = 3/5$ in the right panel.

### 4.2. Superfluid–Insulator Transition of Ultracold Bosons

We consider three independent species of bosonic ultracold atoms, denoted by $(a_{m,n}, b_{m,n}, c_{m,n})$, in a square optical lattice. This system is well-captured by the Bose–Hubbard model and has been theoretically shown to undergo superfluid–insulator transitions. Here we study the effect of the $U(3)$ gauge potentials given in Equation (16) on such transitions, which result in inter-species hopping

terms. Starting from the tight-binding limit, we treat these hopping terms perturbatively. The Hamiltonian of the model is given by

$$H = H_0 + H_1,$$
$$H_0 = \sum_{m,n} \sum_{s=a,b,c} \left[ \frac{\mathcal{U}}{2} \hat{n}^s_{m,n} \left( \hat{n}^s_{m,n} - 1 \right) - \mu \, \hat{n}^s_{m,n} \right],$$
$$H_1 = J \sum_{m,n} \left\{ \Psi^\dagger_{m+1,n} U_x \Psi_{m,n} + \Psi^\dagger_{m,n+1} U_y \Psi_{m,n} \right\} + h.c.,$$
$$\psi^\dagger_{m,n} = \begin{pmatrix} a^\dagger_{m,n} & b^\dagger_{m,n} & c^\dagger_{m,n} \end{pmatrix}, \tag{20}$$

where the interaction strength $\mathcal{U}$ and the chemical potential $\mu$ have been chosen to be the same for all species for simplicity. Here the hopping matrices $U_x$ and $U_y$ are given by Equation (16). We will consider the limit $0 \leq \mu \leq \mathcal{U}$ such that $H_0$ describes three independent species having a unique non-degenerate ground state with $n^s_{m,n} = 1$.

Following the analysis in earlier papers [39–42], the zeroth order Green's function (corresponding to $H_0$) at zero temperature is given by

$$G^0_{s,s'}(\mathbf{k}, \mathbf{k}', i\,\omega) = \delta_{s,s'} \, \delta_{\mathbf{k},\mathbf{k}'} \, G^0(i\,\omega),$$
$$G^0(i\,\omega) = \frac{n_0 + 1}{i\,\omega - E_p} - \frac{n_0}{i\,\omega + E_h},$$
$$E_h = \mu - \mathcal{U}\,(n_0 - 1), \quad E_p = -\mu + \mathcal{U}\,n_0, \tag{21}$$

where $\omega$ is the bosonic Matsubara frequency and $E_h$ ($E_p$) is the energy cost of adding a hole (particle) to the Mott insulating phase. Also, $n_0 = [\mu/\mathcal{U}]$ is the on-site particle number.

The $x$-components of the momenta, in the presence of the flux $\alpha$, are constrained to lie in the magnetic Brillouin zone where two successive points differ by $\pm 2\pi\alpha$. For example, $k_x$ can be assigned the discrete values $2\pi\alpha\,\ell$, where $\ell = 0, 1, \dots Q - 1$. Using this notation, we denote the momentum space wavefunction as $\psi_\ell(\mathbf{k}) \equiv \psi(\mathbf{k} + 2\pi\alpha\,\ell\,\hat{\mathbf{k}}_x)$. The hopping matrix, obtained from $H_1$, is then given by

$$\mathcal{T}_{\mathbf{k},\ell,\mathbf{k}',\ell'} = \delta_{\mathbf{k},\mathbf{k}'} \big[ \mathcal{M}_1(k_x, \ell)\, \delta_{\ell,\ell'} + \mathcal{M}_2(k_y)\, \delta_{\ell+1,\ell'}$$
$$+ \mathcal{M}_2^\dagger(k_y)\, \delta_{\ell-1,\ell'} \big],$$
$$\mathcal{M}_1(k_x, \ell) = J\, e^{i(k_x + 2\pi\alpha\,\ell)}\, U_x + h.c.$$
$$= J \begin{pmatrix} 0 & e^{-i(k_x + 2\pi\alpha\,\ell)} & e^{i(k_x + 2\pi\alpha\,\ell)} \\ e^{i(k_x + 2\pi\alpha\,\ell)} & 0 & e^{-i(k_x + 2\pi\alpha\,\ell)} \\ e^{-i(k_x + 2\pi\alpha\,\ell)} & e^{i(k_x + 2\pi\alpha\,\ell)} & 0 \end{pmatrix},$$
$$\mathcal{M}_2(k_y) = -\frac{J e^{ik_y}}{3} \begin{pmatrix} 1 & 2 & -2 \\ 2 & 1 & 2 \\ -2 & 2 & 1 \end{pmatrix}. \tag{22}$$

The dispersion relations can be found by solving

$$\tilde{\mathcal{M}}_1(k_x, \ell)\, \psi_\ell(\mathbf{k}) - \mathcal{M}_2(k_y)\, \psi_{\ell+1}(\mathbf{k}) - \mathcal{M}_2^\dagger(k_y)\, \psi_{\ell-1}(\mathbf{k}) = 0,$$
$$\tilde{\mathcal{M}}_1(k_x, \ell) = [G^0(\omega_r + i\eta)]^{-1}\, \mathbb{I}_{3\times3} - \mathcal{M}_1(k_x, \ell), \tag{23}$$

where we have analytically continued to real frequencies as $i\,\omega \to \omega_r + i\eta$. In other words, we have to solve the $3Q \times 3Q$ matrix equation

$$\begin{pmatrix} \bar{\mathcal{M}}_1(k_x,0) & -2\,\Re[\mathcal{M}_2(k_y)] & 0 & 0 & 0 & . & . & 0 & 0 & 0 \\ -2\,\Re[\mathcal{M}_2(k_y)] & \bar{\mathcal{M}}_1(k_x,1) & -2\,\Re[\mathcal{M}_2(k_y)] & 0 & 0 & . & . & 0 & 0 & 0 \\ 0 & -2\,\Re[\mathcal{M}_2(k_y)] & \bar{\mathcal{M}}_1(k_x,2) & -2\,\Re[\mathcal{M}_2(k_y)] & 0 & . & . & 0 & 0 & 0 \\ . & . & . & . & . & . & . & . & 0 & 0 \\ . & . & . & . & . & . & . & . & 0 & 0 \\ 0 & 0 & 0 & 0 & 0 & . & . & 0 & -2\,\Re[\mathcal{M}_2(k_y)] & \bar{\mathcal{M}}_1(k_x,Q-1) \end{pmatrix} = 0. \quad (24)$$

The value of the critical hopping parameter $J = J_c$ is obtained when the gap between the lowest particle excitation energy and the highest hole excitation energy goes to zero. The Mott lobes for $\alpha = (0, 1/2)$ are shown in Figure 6.

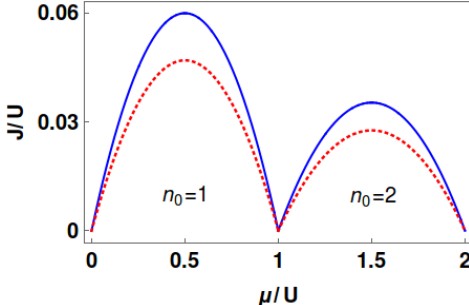

**Figure 6.** The Mott lobes obtained from the critical values of $J/U$. The solid blue (dotted red) curve corresponds to $\alpha = 0\ (1/2)$.

## 5. Discussion

To summarise, we have extended existing studies of ultracold atoms in artificial $U(2)$ gauge potentials to the case of $U(3)$. In doing so, we have considered background gauge fields with both non-constant and constant Wilson loops. We find that the spectrum for the constant Wilson loop case exhibits a fractal structure very similar to the well-studied Abelian case of Hofstader's. Systems with irrational fluxes have been shown to undergo metal–insulator transitions as the hopping parameters are tuned. We have also shown the effect of such a gauge potential in the specific case of the Mott insulator and for superfluid transition for bosonic ultracold atoms subjected to rational flux-values.

There are certain similarities observed with the $U(2)$ cases. For the metal-insulator transition in Section 3.2, the behaviour of the extended/localized states in the $k_y$-$r$ plane are similar to that in the $U(2)$ case [32]. Again, for the superfluid-insulator transition in Section 4.2, the presence of the $U(3)$ flux led to a suppression of the values of $J_c$ with respect to the zero fluz case. Such suppression was also found in the $U(2)$ case [42].

In general, it might be easier to simulate *U(2)* gauge potentials rather than *U(3)* or higher gauge group potentials in cold atom experiments. While systems with *U(2)* gauge potential can be useful to study fermions with the spin degree of freedom, which is what we find in condensed matter systems, the simulation of *U(3)* gauge potentials may open the path to study QCD-like systems.

Our study opens several pathways towards future work involving these systems. For instance, in the fractal case, the Chern numbers for the emerging energy bands can be calculated leading to the identification of the various topological phases. Further, while for the scope of this work, we have limited ourselves to the simplest case of square lattice, it will be interesting to study cases with other structures such as triangular and honeycomb lattices. Future exploration along these directions will give a better theoretical understanding of such systems. It will also help in optimising design related decisions for experiments in the field and suggest the experimental signatures one ought to go hunting for.

**Acknowledgments:** I.M. is supported by NSERC of Canada and the Templeton Foundation. A.B. is supported by the Fonds de la Recherche Scientifique-FNRS under grant number 4.4501.15. In addition A.B. is grateful for

the hospitality provided by the Perimeter Institute during the completion of the work. Research at the Perimeter Institute is supported, in part, by the Government of Canada through Industry Canada and by the Province of Ontario through the Ministry of Research and Information.

**Author Contributions:** I.M. conceived and set up the problem and the equations; A.B. wrote numerical codes for solving the equations and plotting the data obtained.

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
