# Peer review of "Cold Atoms in U(3) Gauge Potentials"

_condensedmatter, doi:10.3390/condmat1010002_

Reviewer 1 Report

The paper studies ultracold atoms in the the presence of an  artificial U(3 gauge potentials. Artificial gauge fields in the field of ultracold atoms is a rapidly developing field of very high interest. I think this paper will be interest to readers and support its publication.

Author Response

Thanks for your positive cmments.

Reviewer 2 Report

By comparing the gauge fields with non-constant and constant Wilson loops, the authors studied  the effects of artificial U(3) gauge potentials on ultracold atoms. The paper reads well and the theoretical model is clearly discussed. Therefore, I would recommend the manuscript for publication in Condensed Matter after the authors make the minor modifications.

(1) Please rewrite the figure caption of Figure 5 to make the statement of right figure clear.

(2) The authors should put more sentences in conclusion of how U(2) gauge potentials is relevant to artificial U(3) gauge potentials on ultracold atoms.

Author Response

1) The caption has been changed to:

"Energy spectrum from Eq.~(\ref{eigqn2}) for $r= 1$ as a function of (i) $\alpha$ in the left panel, and (ii) $k_y$ at $\alpha= 3/5$ in the right panel."

2) We added the following lines in the consluion section:

<< There are certain similarities observed with the $U(2)$ cases. For the metal-insulator transition in Sec.~\ref{trans1},  the behaviour of the extended/localized states in the $k_y$-$r$ plane are similar to that in the $U(2)$ case \cite{Clark}. Again, for the superfluid-insulator transition in Sec.~\ref{trans2}, the presence of the $U(3)$ flux led to a suppression of the values of $J_c$ with respect to the zero fluz case. Such suppression was also found in the $U(2)$ case \cite{krish-kush}.

In general, it might be easier to simulate U(2) gauge potentials rather than U(3) or higher gauge group potentials in cold atom experiments. While systems with U(2) gauge potential can be useful to study fermions with the spin degree of freedom, which is what we find in condensed matter systems, the simulation of U(3) gauge potentials may open the path to  study QCD-like systems.>>

Round  2

Reviewer 2 Report

I am agree with the manuscript for publication in Condensed Matter.